# Development of a Multipurpose Core Collection of New Promising Iranian Pomegranate (*Punica granatum* L.) Genotypes Based on Morphological and Pomological Traits

**Sara Razi** [1,2], **Ali Soleimani** [1], **Mehrshad Zeinalabedini** [2,*,†], **Mohammad Reza Vazifeshenas** [3], **Pedro Martínez-Gómez** [4], **Asghar Mohsenzade Kermani** [5], **Ahmad Reza Raiszadeh** [5], **Mostafa Tayari** [5] and **Pedro José Martínez-García** [4,*,†]

1   Department of Horticulture, Faculty of Agriculture, University of Zanjan, Zanjan 45371-38791, Iran; Sara.razi86@gmail.com (S.R.); asoleimani@znu.ac.ir (A.S.)
2   Department of Systems and Synthetic Biology, Agricultural Biotechnology Research Institute of Iran (ABRII), Agricultural Research, Education and Extension Organization (AREEO), Karaj 31359-33151, Iran
3   Agriculture and Natural Resources Research Center of Yazd, Agricultural Research, Education and Extension Organization (AREEO), Yazd 89165-571, Iran; asman2000@gmail.com
4   Department of Plant Breeding, CEBAS-CSIC, P.O. Box 164, Espinardo, E-30100 Murcia, Spain; pmartinez@cebas.csic.es
5   Department of Plant Production, Organization Agriculture, Isfahan 81746-79611, Iran; m.b.e@agri-es.ir (A.M.K.); reiszade@agri-es.ir (A.R.R.); tayari1339@yahoo.com (M.T.)
*   Correspondence: m_zeinalabedini@yahoo.com (M.Z.); pjmartinez@cebas.csic.es (P.J.M.-G.)
†   These authors are contributed equally to this work.

**Abstract:** Establishment of a core collection, of limited size and better representation of the whole germplasm phenotypic diversity, is fundamental for fruit tree breeding programs from an economic and management points of view. To achieve this goal with pomegranate fruit trees, 221 genotypes were evaluated for 25 morphological and pomological traits during two successive years. Using the maximization strategy in Power Core software, 12 out of 221 pomegranate genotypes were selected for the new core collection, reducing the population size to 5.42% of the entire collection. Variance difference (VD%) and mean difference (MD%) were calculated as 42.68% and 7.03% in core collection, respectively. This indicates an excellent diversity amongst genotypes within the core collection. The Shannon's diversity index ($H'$) in the formed collection suggested that 19 out of 25 phenotypic variables were of high diversity. Results showed that core collection genotypes are equally presented in all three population groups formed by cluster analysis through the original collection. The current research is the first in using phenotypic data to establish a core collection of Iranian pomegranate germplasm. The formation of this core collection will be an effective step towards examining the diversity of the original population and breeding prospects.

**Keywords:** genetic resource; phenotypic diversity; germplasm conservation; breeding program

## 1. Introduction

Pomegranate (*Punica granatum* L.) is native to central Asia, particularly some areas of Iran and that was broadcasted to other parts of the world [1–4]. Extensive cultivation of pomegranate in tropical and subtropical regions in a new changing climatic context indicates the high adaptability and flexibility of this species [5]. Pomegranate fruits are characterized by their high antioxidant properties and phenolic contents, which can amend metabolic syndrome [6,7]. With an annual production of more than 1 million tons and a cultivated surface of 89,400 ha, Iran is one of the largest pomegranate producers in the world [8].

Phenotypic variability in this tree species is the result of the creation of none "true-to-type" seedlings of pomegranate due to about 13% outcrossing [9] and morphological

changes created through the domestication process. The genetic and phenotypic base of these variations in Iranian pomegranate germplasm has been studied using morphological characteristics [10,11] and molecular features [12]. Accordingly, various studies have been conducted worldwide on the biological diversity of pomegranate to collect, preserve, and evaluate its germplasm [13–15].

Despite the large number of pomegranate accessions with different phenotypic characteristics throughout the world, only a few are commercially acceptable and widely cultivated. Therefore, due to the selection of superior accessions for breeding purposes as well as orchard establishment, most of the existing genetic diversity was not included in the commercial cultivation of this fruit species [16]. The pomegranate breeding infrastructure includes highly variable pomegranate collection [17], segregating populations for important morphological and pomological features, assembled transcriptome, SNP markers, and a genetic map [18]. Fruit related characteristics have high potential for discrimination among different pomegranate accessions [15]. Traits with commercial significance in pomegranate which have the most goals in its breeding program are included tree yield, fruit skin color, aril color, fruit size and shape, seed softness, active pharmaceutical compounds, and resistance to fruit cracking and sunburn [16,19].

Classifying genetic resources for the identification of superior genotypes is of utmost importance for the successful design of breeding programs, and has a significant role in facilitating the management of genetic collections' conservation [15]. Gene banks can provide researchers with genetic resources for long-term protection of plants, mainly perennial fruit trees, and rich germplasm sources. Still, the formation of a core collection, more limited in size and more representative of the genetic diversity of the whole gene bank, is an effective approach in terms of spending less time and money in the production and breeding context [20,21].

Formation of most core collections based on the so-called maximizing method using the MSTRAT software is done by optimizing the number of traits for germplasm protection, while the number of accessions and diversity in the collection determines the size of the core [22]. Genetic diversity and variability determine sample size, which cannot be a uniform standard because of different crop species' number and the creation of specific characteristics through evolution and human intervention [20]. The core collection size varies from 5 to 30% of the entire population [23,24]. According to Brown's [25] suggestion, a sample size of 5 to 10% of the entire collection with a maximum of 3000 samples per species can preserve about 70% of the alleles of the entire collection. Over the past decade, a core collection has been created in several perennial fruit tree species, such as apricot [20,26], olive [27], apple [28], and walnut [29].

To the best of the researcher's knowledge, no study or guideline exists on establishing a core collection for pomegranate fruit trees based on neither molecular nor phenotypic traits. Hence, the present study aimed to create a multipurpose core collection based on the evaluation of different morphological and pomological attributes. It should contain the diversity present in the Iranian National Pomegranate Collection (INPC) so that while keeping the most important phenotypic traits, superior genotypes are identified.

## 2. Materials and Methods

### 2.1. Plant Material

The INPC held in Yazd Province was used as a plant material supplier for this study. Since 1987, more than 760 new promising pomegranate genotypes from the whole country have been collected in Yazd pomegranate collection in Agriculture and Natural Resources Research Centre of Yazd (31°55′ N, 54°16′ E, and 1216 m alt.). Trees in this collection were collected from different regions of Iran and asexually propagated. To represent as much of the diversity of the germplasm, 221 out of more than 760 genotypes of this collection were selected based on preliminary phenotype characterizations due to its superior behavior. Supplementary Table S1 indicates that most samples are originated from

the central provinces of Iran and provides the genotype accession codes and names. The accession codes are based on the province code and sample number in the Yazd collection.

### 2.2. Agro-Morphological Evaluation

A total of 25 traits related to tree appearance, leaf, flower, and fruit were evaluated during two successive years (2012–2013) (The average monthly air temperature for the growing season = 28.2 °C, average annual values = 20.8 °C and average amount of precipitation = 27 mm for those two years) according to international descriptor of International Union for the Protection of New Varieties of Plants [30] by obtaining 30 samples from each accession. Evaluated traits as designated with their respective codes and categories are presented in Supplementary Table S2.

### 2.3. Establishment of the Multipurpose Core Collection

Development of the core collection was performed based on analysis of the morphological attributes, using the maximization strategy in the PowerCore V.1.0 software as proposed by Kim et al. [31]. In order to evaluate superior genotypes that will constitute the core germplasm, four parameters were evaluated including variance difference percentage (VD%), mean difference percentage (MD%), changeable rate of the coefficient of variation (VR%), and coincidence rate of range (CR%).

### 2.4. Statistical Analysis

Descriptive statistics were carried out for both the entire and core collections. Morphological diversity for each morphological and pomological trait was estimated using the $H'$ index according to Shannon [32]:

$$H' = -\sum_{n=1}^{n} Pi \times ln(Pi)$$

where $H'$ is the diversity index, $Pi$ is the proportion of each phenotypic trait in the sample, and $ln(Pi)$ is the natural logarithm of this proportion. In factor analysis and according to construction factorial coefficients matrix, the factors with eigenvalues greater than 1.0 were evaluated [33]. The percentage of explained variability for each factor and the communalities for each trait were determined, and the scatterplots were established based on two first factors. Besides, the non-parametric Spearman correlation approach was used to assess the correlation between traits with a breeding purpose. The genetic dissimilarity component was analyzed using Euclidean distance, and the agglomerative hierarchical clustering algorithm was performed by Ward's method. Statistical analysis and drawing the plots were performed by R software 4.0.4 and XLSTAT software (Version 2016.1).

## 3. Results

### 3.1. Description of Morphological and Pomological Traits

The observations on 25 traits were recorded for all 221 genotypes. Data analysis is presented in Supplementary Table S2 and Figure 1. Based on frequency distribution, a high variability was recorded for the number of fruits per tree, fruit mean weight and thorn in mature wood branch, fruit size, fruit skin thickness, fruit symmetry, and aril color. In contrast, tree crown shape, length-to-width leaf ratio, fruit skin sensitivity to sunburn, fruit crown shape, aril size, and seed hardness showed less variability. The descriptive statistics of minima, maxima, mode, median, and $H'$ are presented in Table 1. Modified from Eticha et al. [34] and Jamago and Cortes [35], an ideal classification of high ($H' \geq 0.67$), intermediate ($0.34 \leq H' \leq 0.66$), or low ($0.01 \leq H' \leq 0.33$) diversity indices was adapted. While a high diversity was found in terms of some pomological attributes such as FSY (1.13), TMY (1.06), and NFT (1), others such as TCS (0.43), FCD (0.55), and FST (0.48) showed intermediate, or such as SH (0.05), FSSS (0.15), and AS (0.18) showed low variability among the studied genotypes. The diversity index is presented in Table 1.

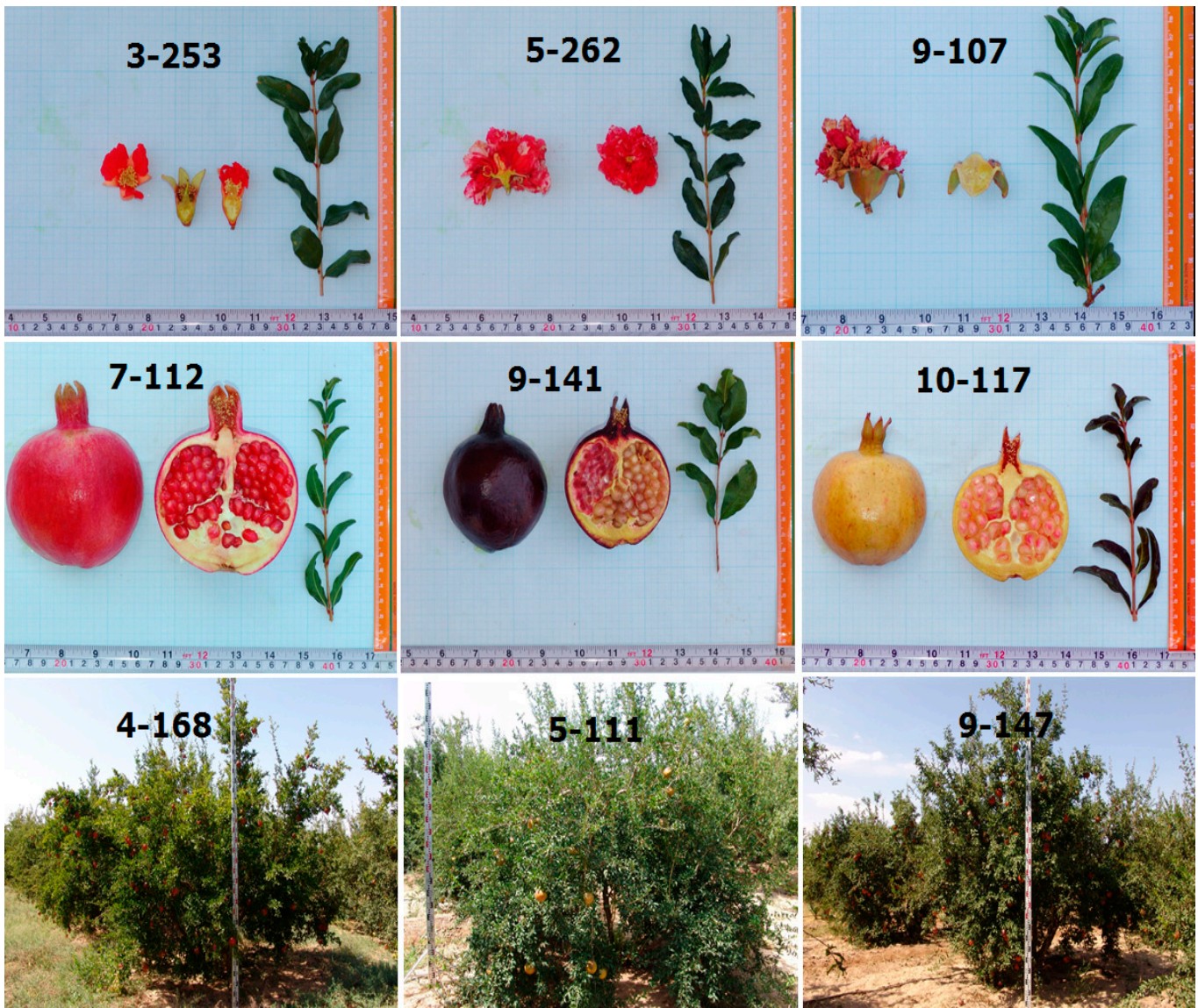

**Figure 1.** Evaluation of flower, fruit, and tree traits among different Iranian pomegranate genotypes.

Most pomegranate genotypes showed a wide tree crown shape, a strong tree growth power, and a medium length-to-width leaf ratio. In addition, most of the genotypes formed flowers on the yearly branches with lateral positions and medium cup diameters. The high fruit skin sensitivity to burst, medium fruit skin sensitivity to sunburn, and medium fruit skin thickness were the most common. Also, most of the genotypes had medium fruit size, sour-sweet flavor, medium aril size, and light red aril color with hard seeds (Tables 1 and S2).

### 3.2. Correlations between Morphological and Pomological Traits

Flower position showed a positive correlation with fruitful flower percentage, fruit symmetry, and fruit crown shape and a negative correlation with dominant flowering habit and fruit shape. A positive relationship was observed between flower cup diameter and productivity traits such as mean tree yield, mean fruit weight, and number of fruits per tree. There was a negative relationship between length-to-width leaf ratio with fruit skin sensitivity to sunburn and burst. In addition, it was found that the fruit skin sensitivity to burst had a positive relationship with the fruit skin sensitivity to sunburn and a negative correlation with fruit skin thickness (Figure 2).

**Table 1.** Descriptive statistics of the phenotypic traits in both entire and core collections of 221 pomegranate genotypes. Tree Crown Shape (**TCS**); Tree Growth Power (**TGP**); Thorn in Mature Wood Branch (**TMB**); Length−to−width Leaf Ratio (**LWLR**); Flower Position (**FP**); Flower Formation Site (**FFS**); Dominant Flowering Habit (**DFH**); Fruitful Flowers Percentage (**FFP**); Flower Cup Diameter (**FCD**); Fruit Size (**FS**); Fruit Skin thickness (**FST**); Fruity Skin Sensitivity to Burst (**FSSB**); Fruit Skin Sensitivity to Sunburn (**FSSS**); Fruit Flavor (**FF**); Fruit Ripening Time (**FRT**); Fruit Skin Color (**FSC**); Fruit Shape (**FSH**); Fruit Crown Shape (**FCS**); Fruit symmetry (**FSY**); Aril Size (**AS**); Seed Color (**SC**); Seed Hardness (**SH**); Number of fruits in tree (**NFT**); Fruit mean weight (**FMW**); Tree mean yield (**TMY**).

| Trait | Core Collection | | | | | Entire Collection | | | | |
|---|---|---|---|---|---|---|---|---|---|---|
| | **Min** | **Max** | **Mode** | **Median** | **Heterozygosity** | **Min** | **Max** | **Mode** | **Median** | **Heterozygosity** |
| TCS | 2 | 3 | 3 | 3 | 0.64 | 2 | 3 | 3 | 3 | 0.43 |
| TGP | 1 | 3 | 3 | 3 | 0.82 | 1 | 3 | 3 | 3 | 0.95 |
| TMB | 1 | 7 | 1 | 1 | 0.98 | 1 | 7 | 1 | 1 | 0.83 |
| LWLR | 5 | 7 | 5 | 5 | 0.68 | 5 | 7 | 5 | 5 | 0.69 |
| FP | 1 | 5 | 1 | 3 | 1.07 | 1 | 5 | 1 | 1 | 0.92 |
| FFS | 1 | 2 | 1 | 1 | 0.64 | 1 | 2 | 1 | 1 | 0.64 |
| DFH | 1 | 2 | 1 | 1.5 | 0.69 | 1 | 2 | 1 | 1 | 0.69 |
| FFP | 1 | 2 | 2 | 2 | 0.68 | 1 | 2 | 1 | 1 | 0.68 |
| FCD | 3 | 7 | 5 | 5 | 1.03 | 3 | 7 | 5 | 5 | 0.55 |
| FS | 3 | 9 | 5 | 5 | 0.98 | 3 | 9 | 5 | 5 | 0.26 |
| FST | 1 | 7 | 3 | 3 | 0.84 | 1 | 7 | 3 | 3 | 0.48 |
| FSSB | 3 | 7 | 7 | 7 | 0.72 | 3 | 7 | 7 | 7 | 0.66 |
| FSSS | 3 | 5 | 5 | 5 | 0.29 | 3 | 5 | 5 | 5 | 0.15 |
| FF | 1 | 3 | 1 | 2 | 1.02 | 1 | 3 | 2 | 2 | 0.84 |
| FRT | 3 | 7 | 5 | 5 | 0.57 | 3 | 7 | 5 | 5 | 0.26 |
| FSC | 3 | 6 | 3 | 4 | 1.01 | 3 | 6 | 3 | 3 | 0.94 |
| FSH | 1 | 3 | 1 | 2 | 1.02 | 1 | 3 | 2 | 2 | 0.95 |
| FCS | 1 | 2 | 2 | 1.5 | 0.69 | 1 | 2 | 2 | 2 | 0.69 |
| FSY | 2 | 5 | 4 | 4 | 1.31 | 2 | 5 | 4 | 4 | 1.13 |
| AS | 1 | 3 | 3 | 3 | 0.45 | 1 | 3 | 3 | 3 | 0.18 |
| AC | 1 | 4 | 3 | 3 | 0.84 | 1 | 4 | 3 | 3 | 0.19 |
| SH | 1 | 5 | 5 | 5 | 0.29 | 1 | 5 | 5 | 5 | 0.05 |
| NFT | 1 | 9 | 9 | 7 | 1.44 | 1 | 9 | 7 | 7 | 1 |
| FMW | 1 | 9 | 3 | 3 | 1.23 | 1 | 9 | 3 | 3 | 0.7 |
| TMY | 1 | 5 | 1 | 2 | 1.01 | 1 | 5 | 3 | 3 | 1.06 |

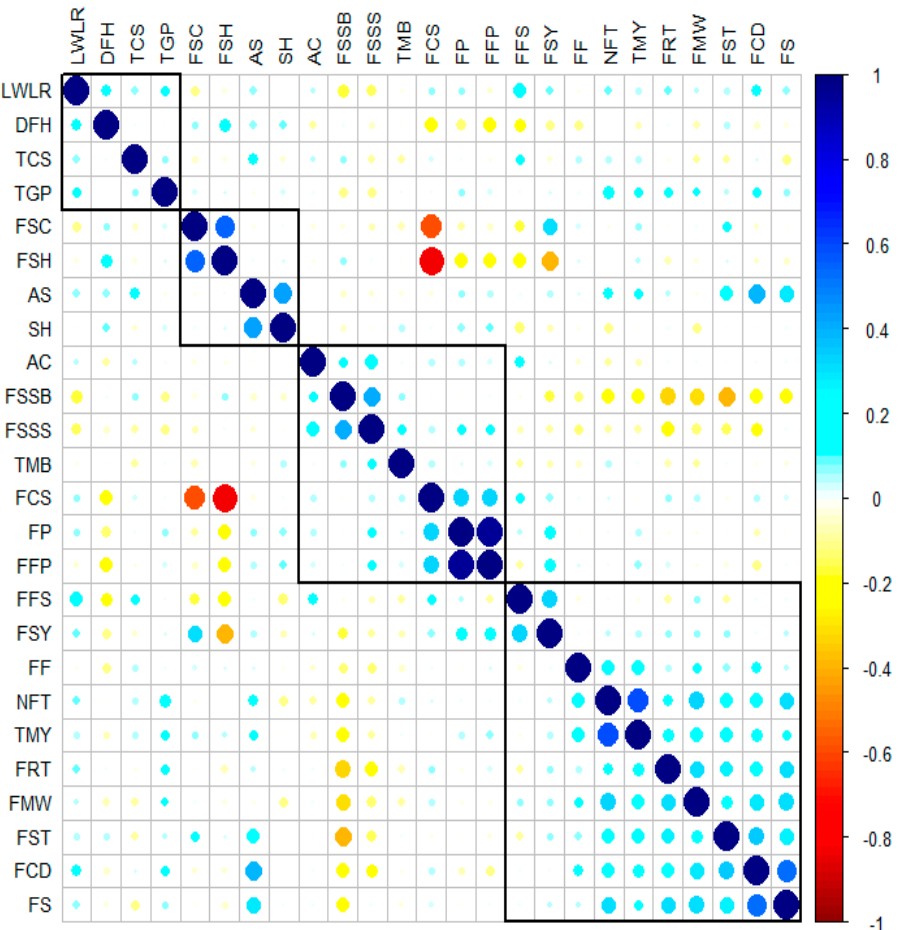

**Figure 2.** Spearman correlation coefficients among 25 morphological and pomological traits of 221 pomegranate genotypes. Tree Crown Shape (**TCS**); Tree Growth Power (**TGP**); Thorn in Mature wood Branch (**TMB**); Length-to-width Leaf Ratio (**LWLR**); Flower Position (**FP**); Flower Formation Site (**FFS**); Dominant Flowering Habit (**DFH**); Fruitful Flowers Percentage (**FFP**); Flower Cup Diameter (**FCD**). Fruit Size (**FS**); Fruit Skin thickness (**FST**); Fruity Skin Sensitivity to Burst (**FSSB**); Fruit Skin Sensitivity to Sunburn (**FSSS**); Fruit Flavor (**FF**); Fruit Ripening Time (**FRT**); Fruit Skin Color (**FSC**); Fruit Shape (**FSH**); Fruit Crown Shape (**FCS**); Fruit symmetry (**FSY**); Aril Size (**AS**); Seed Color (**SC**); Seed Hardness (**SH**); Number of fruits in tree (**NFT**); Fruit mean weight (g) (**FMW**); Tree mean yield (**TMY**).

The classification of the 25 evaluated traits divided the traits into four separate groups. In the first group, tree-related attributes LWLR, TCS, TGP, and DFH were included. AS, SH, FSH, and FSC were in the second group. FP, FFP, and five traits related to fruit established the third group. In the fourth group, there were ten traits that were mainly related to fruit yield and appearance (Figure 2).

*3.3. Multivariate Analysis*

The results of the factor analysis using the Varimax Rotation method [36] allowed the classification of the 25 morphological traits in seven main groups, which justified and covered 76.53% of the total variance. The first and second factors each justified 25.41 and 11.93% respectively, and 37.34% of the cumulative variance. Examining the relationship between traits and factors showed that fruit skin color, shape, and crown shape are most related to the first factor (Figure 3A), and flower position and fruitful flower percentage were dominant in the second factor (Figure 3B). The third factor (11.24%) consisted of traits including fruit skin sensitivity to burst and sunburn, fruit skin thickness, and fruit ripening time. The fourth to seventh factors explained 9.26, 7.79, 5.88, and 4.99% of the total variance, respectively.

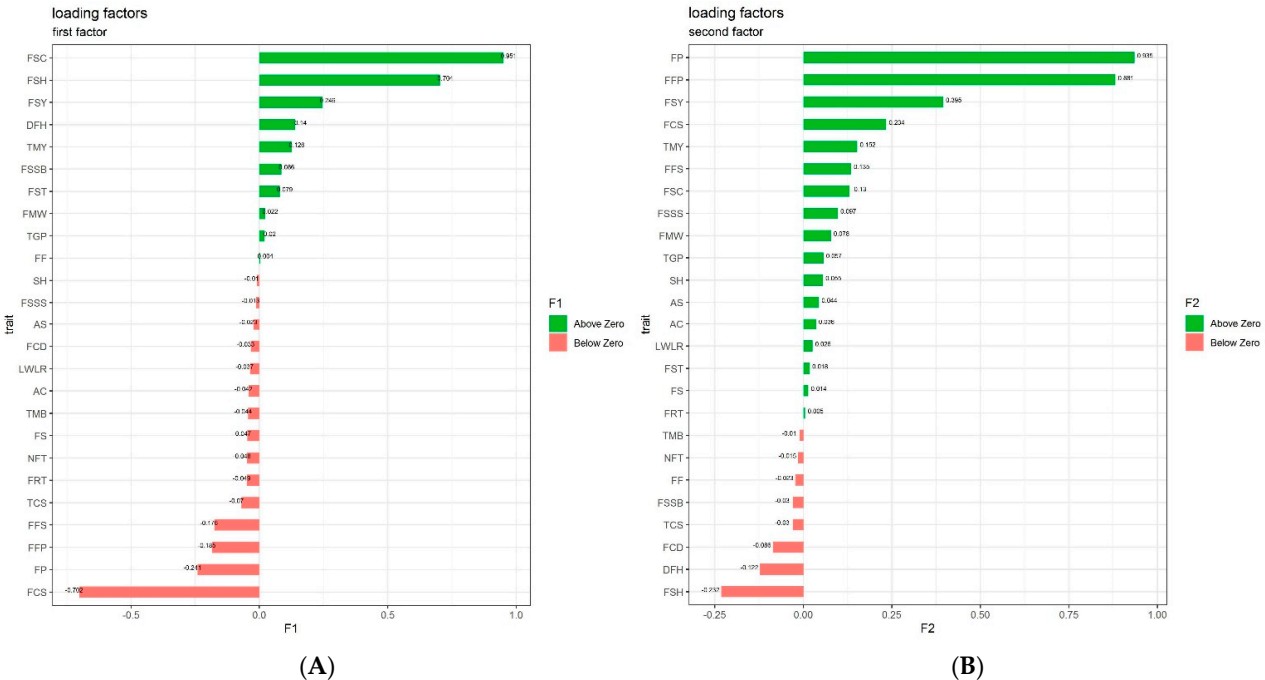

(**A**)                                                                                          (**B**)

**Figure 3.** Eigenvectors for the variables of the first (**A**) and second (**B**) factors. Tree Crown Shape (**TCS**); Tree Growth Power (**TGP**); Thorn in Mature wood Branch (**TMB**); Length−to−width Leaf Ratio (**LWLR**); Flower Position (**FP**); Flower Formation Site (**FFS**); Dominant Flowering Habit (**DFH**); Fruitful Flowers Percentage (**FFP**); Flower Cup Diameter (**FCD**). Fruit Size (**FS**); Fruit Skin thickness (**FST**); Fruity Skin Sensitivity to Burst (**FSSB**); Fruit Skin Sensitivity to Sunburn (**FSSS**); Fruit Flavor (**FF**); Fruit Ripening Time (**FRT**); Fruit Skin Color (**FSC**); Fruit Shape (**FSH**); Fruit Crown Shape (**FCS**); Fruit symmetry (**FSY**); Aril Size (**AS**); Seed Color (**SC**); Seed Hardness (**SH**); Number of fruits in tree (**NFT**); Fruit mean weight (g) (**FMW**); Tree mean yield (**TMY**).

A two-dimensional scatter plot was drawn based on the first and second factors, and genotypes were grouped from 1 to 10 groups (Figure 4).

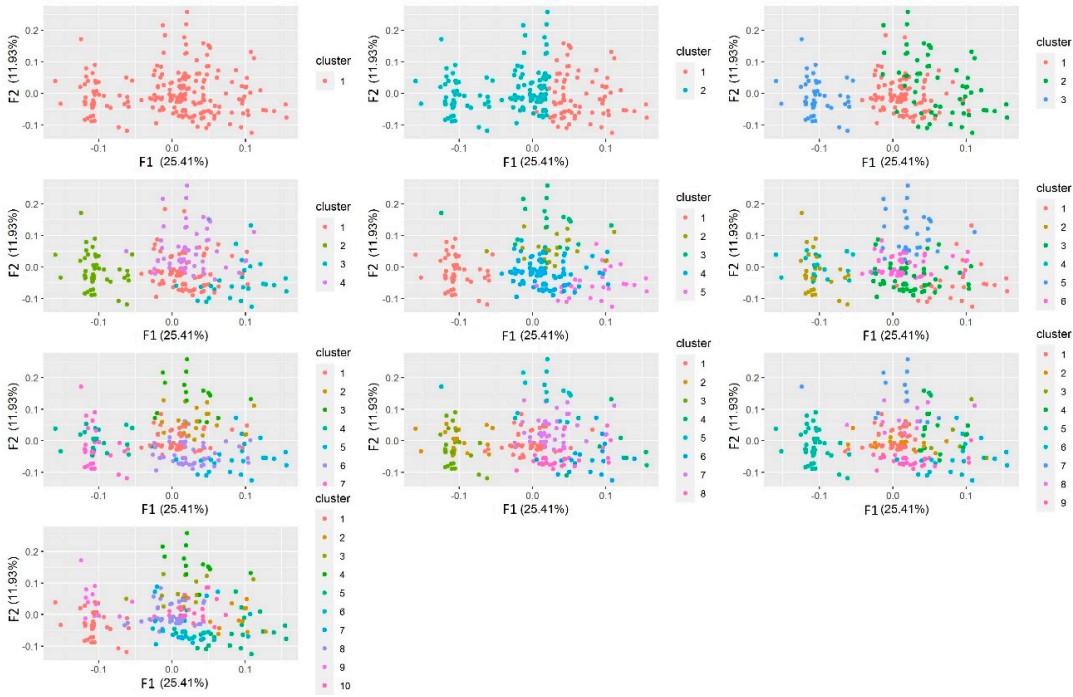

**Figure 4.** Scatter plot of the distribution of 221 pomegranate genotypes based on the first two factors from k = 1 to k = 10.

According to k-mean partitions comparison analysis based on Calinski criterion, K = 3 was selected as the best k (Figure 5A), and 221 genotypes were divided into three main groups (Figure 5B).

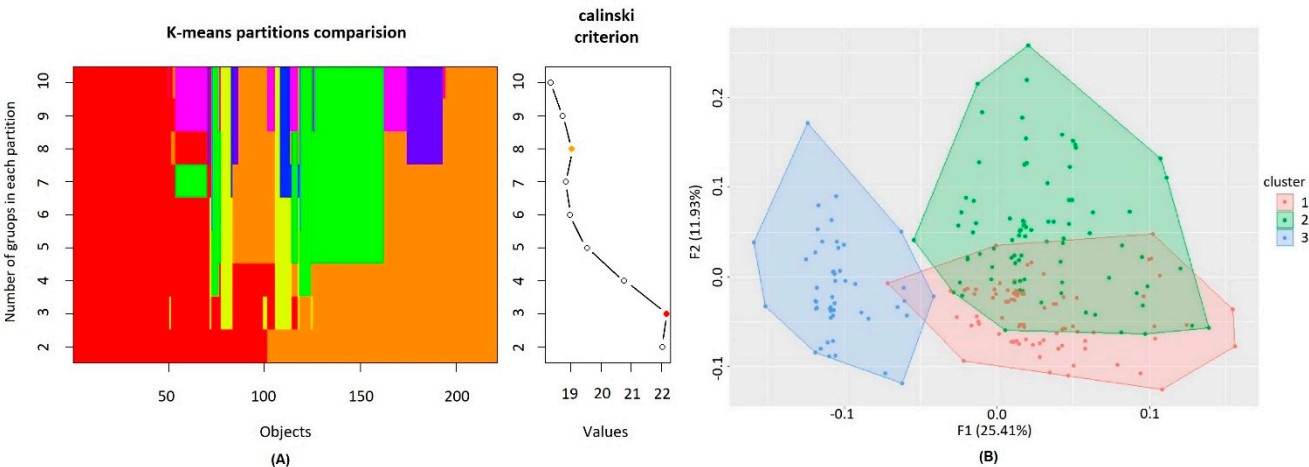

**Figure 5.** K − mean analysis to determine the best k for grouping 221 pomegranate genotypes based on scatter plot (**A**) and scatter plot showing the distribution of evaluated genotypes based on k = 3, using the two first factors (**B**).

On the other hand, consistent with the results of K analysis, the cluster analysis of the assayed genotypes, using the ward's method and Euclidean distance, revealed three main groups. The first (C1) and the second (C2) groups consisted of 26 and 60 genotypes, respectively. The remaining 135 genotypes were classified in the third group (C3) with two sub-clusters i.e., C3a and C3b, consisting of 10 and 125 individuals, respectively (Figure 6).

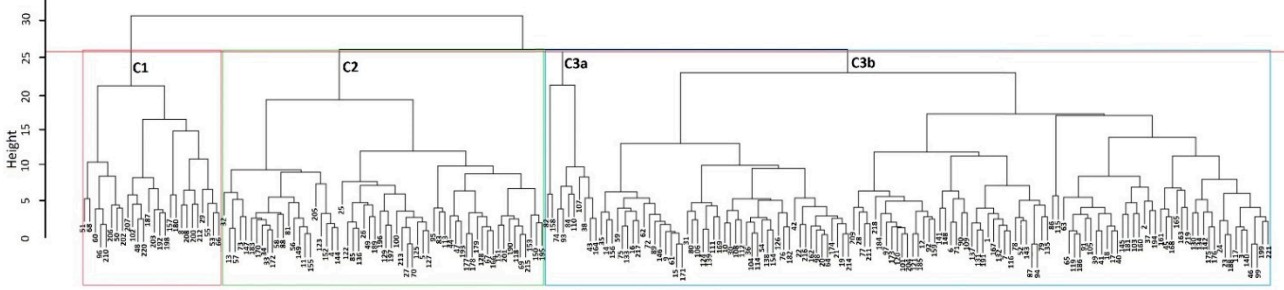

**Figure 6.** Ward cluster analysis of the 221 pomegranate genotypes based on morphological and pomological traits using Euclidean distances.

The relationship between traits and each group of genotypes was determined. Based on these results, fruit shape, fruit skin color, and dominant flowering habit had the most, and fruit crown shape had the least with the first group. Fruit ripening time, fruit skin thickness, fruit size, flower cup diameter, fruit mean weight, fruit flavor, tree mean yield, and number of fruits per tree showed the most association with the second group. The traits most related to the third group were fruit skin sensitivity to sunburn and burst, flower position, fruitful flower percentage, and fruit crown shape (Figure 7).

*3.4. Construction of the Core Collection*

Using the maximization strategy used in Power Core software, 12 out of 221 pomegranate individuals were selected in the core collection, reducing population size to 5.42% for the entire collection. The formed core collection consisted of the genotypes namely, " Sheitooni-Seyedon", "Robab-Ghasrdasht", "Shahvar-Estahban", "Shahvar-poostghermez", "Danehsiah-Dastjerd",

"Sabi-Bam", "Bitalaf-Daneh-Sefid", "Malisak-Hoshak", "Poost-Ghermez-Bazman", "Bihasteh-Sangan", "Bazmani-Poost-Koloft", and "Ghermez-Harabarjan" (Table S1).

The number of accessions for each class, the core count, and the entire count for all 25 studied morphological and pomological traits are presented in Figure 8. Statistical indices and Shannon's index evaluated the efficiency and variation of candidate core collection (Table 1). In order to evaluate the efficiency of candidate core collections, four traits including MD%, VD%, CR%, and VR% were calculated as 7.03, 42.68, 100, and 150.2%, respectively.

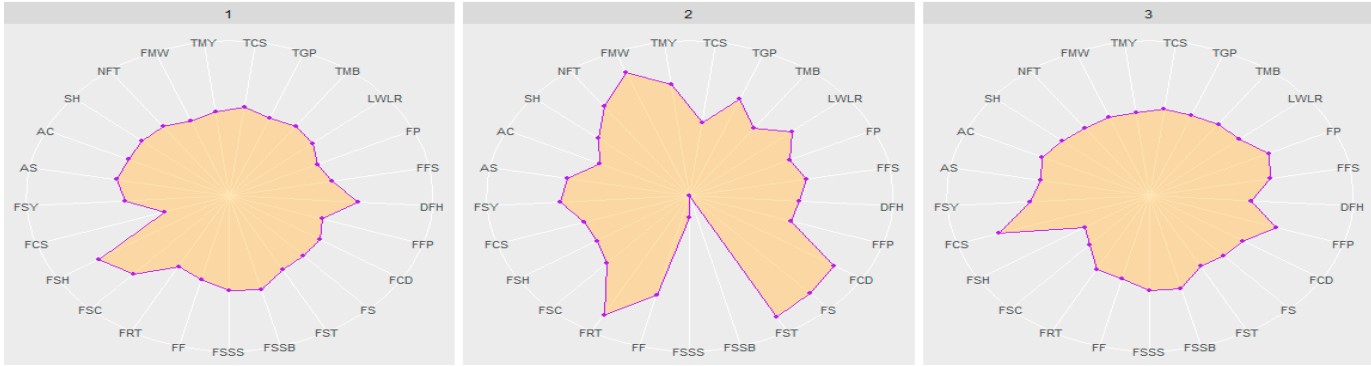

**Figure 7.** Relationships between pomegranate morphological traits and different groups resulted from cluster analysis. Tree Crown Shape (**TCS**); Tree Growth Power (**TGP**); Thorn in Mature wood Branch (**TMB**); Length-to-width Leaf Ratio (**LWLR**); Flower Position (**FP**); Flower Formation Site (**FFS**); Dominant Flowering Habit (**DFH**); Fruitful Flowers Percentage (**FFP**); Flower Cup Diameter (**FCD**). Fruit Size (**FS**); Fruit Skin thickness (**FST**); Fruity Skin Sensitivity to Burst (**FSSB**); Fruit Skin Sensitivity to Sunburn (**FSSS**); Fruit Flavor (**FF**); Fruit Ripening Time (**FRT**); Fruit Skin Color (**FSC**); Fruit Shape (**FSH**); Fruit Crown Shape (**FCS**); Fruit symmetry (**FSY**); Aril Size (**AS**); Seed Color (**SC**); Seed Hardness (**SH**); Number of fruits in tree (**NFT**); Fruit mean weight (g) (**FMW**); Tree mean yield (**TMY**).

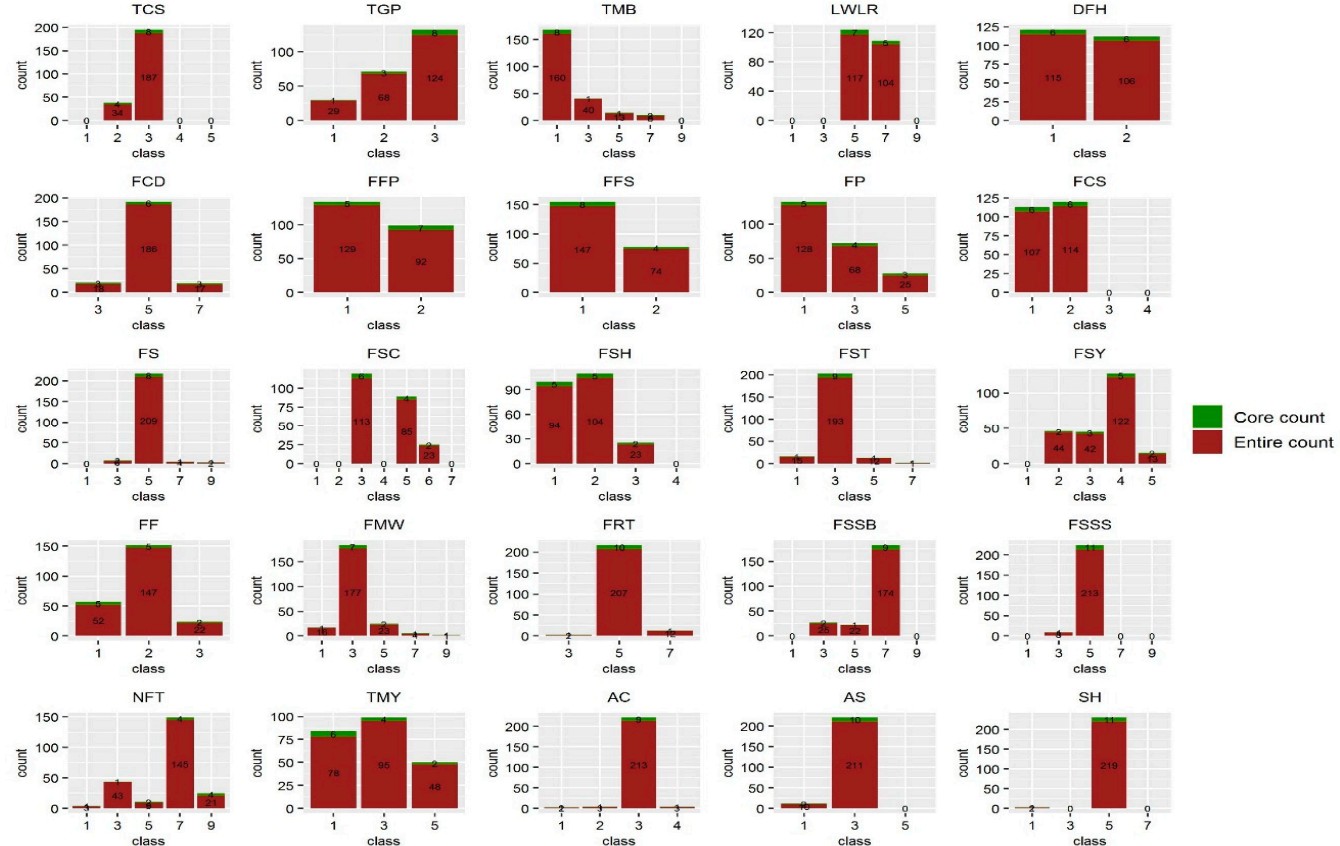

**Figure 8.** Frequency distribution of evaluated pomegranate genotypes in primary and core collections.

## 4. Discussion

A core collection can be formed based on different molecular, morphological, and agronomic markers. The morphological data are usually applied extensively as they are recorded comprehensively [37,38]. The PowerCore software selects genotypes with higher diversity, representing trait states and total coverage of marker alleles of the entire collection through a modified heuristic algorithm [39]. As the formed core collection must be validated, one of the ways to measure and evaluate the formed core collection is to compare its diversity with the original collection's diversity. Therefore, study of the entire collection in terms of morphological and agronomical diversity of existing genotypes and their relationships and grouping is inevitable.

There existed a positive relationship between length-to-width leaf ratio and flower cup diameter, dominant flowering habit, and flower formation site. This means that increased leaf area makes flowering more regular and on perennial branches with larger flowers. Accordingly, the superior genotype can be selected for breeding at the seedling stage through the calculation of leaf area before reaching the maturity and fruiting stage. Regarding the negative relationship of fruit skin sensitivity to sunburn and burst with length-to-width leaf ratio, it should be noted that leaves provide shade and prevent direct sunlight falling on fruits. Hence, growing cultivars with more leaf areas in places where the maximum mean temperature is high can reduce incidences of sunburn [16]. As the length-to-width leaf ratio increases, the leaf area and consequently, the transpiration rate increase. Leaf transpiration rate is known to affect calcium and boron absorption and translocation through fruit peel, which is related to diversity of fruit cracking among cultivars [16,40]. In this regard, Rodriguez et al. [41] reported that anti-transpiration treatments decreased stomatal conductance and resulted in more sunburned fruit in treated trees.

A positive relationship between fruit skin sensitivity to burst and sunburn should be due to the fact that free radicals formed by sunburn destroy the integrity of cell fats, proteins and membrane, which results in cracking [42]. A similar result in pomegranate has been reported by Shakeri and Sadat Akhavi [43]. The relationship between flower position and other traits showed that terminal flowering genotypes have a more regular flowering habit with a higher fruitful flower percentage, and the formed fruit will be more symmetrical and spherical with shorter crowns. In pomegranate, lateral flowers have been reported to have fewer and no developed ovules that are less fertile [44]. Similar results have been achieved in this research, as flowers with large cup diameter were accompanied with high tree productivity.

Examining the relationship between different morphological and pomological evaluated traits through factor analysis, the redundancy among variables was reduced using a smaller number of factors. The first factor, namely "fruit appearance", consisted of traits related to fruit shape such as FSH, FSC, and FCS. In the same vein, FP and FFP were dominant traits in the second factor, namely "flower attributes". The "physiological disorder" was the third factor in which FSSS, FSSB, and FST were representatives. These attributes were more effective in distinguishing and identifying the studied population. Besides, they could be useful for selecting superior genotypes or new cultivars from an economic point of view. In accordance with these results, it has been reported that fruit-related traits are more important in analyzing and differentiating breeding materials, and dealing with the phenotypic characterization of cultivated pomegranates [45,46].

Based on the scatter plot obtained using the first and second factors, genotypes were divided into three groups without any special geographical relationships. The uniform distribution in an area, independent from the geographical origin, has been reported in the grouping of pomegranate genotypes of China [47], Iran [12], Tunisia [48], and India [49] based on genetic and phenotypic data. Accordingly, there was no specific relationship between samples' placement into the groups and provincial divisions in the cluster analysis approach. One of the reasons for this dispersion should be the lack of knowledge on the exact origin of the genotypes, and each may have been given different names in different geographical areas by indigenous people as in the cases of Lebanese [14] and

Turkish pomegranate genotypes [13]. Mislabeling was also reported for pomegranate accessions growing in Tunisia [50]. They demonstrated that the genetic distance between genotypes is rarely correlated with their geographical origins, and the same planting material, depending on the cultivation area, may have different denominations. Two important seedless genotypes, namely 11–223 and 5–220, and several wild genotypes were classified together in the C3a cluster. Most C1 members were commercial genotypes with higher fruit quality attributes and more tolerance to cold stress [51].

Fruit shape, fruit skin color, and dominant flowering habit were most associated with this group. The C2 genotypes were often cold-sensitive, and most of their fruits are used for processing purposes. Productivity properties, such as fruit size and mean weight, number of fruits in tree, tree mean yield and also flower cup diameter, fruit ripening time, and fruit skin thickness were the traits associated with this group. Fruitful flower percentage, flower position, fruit crown shape, and skin sensitivity to sunburn and burst were mostly associated with C3.

After the establishment of a core collection, an important issue is the extent to which it meets its original objectives in terms of the lack of repetition and diversity representation. In this concept, comparison of the core collection with its original one is usually used for validation purposes. Examination of the formed core collection revealed that the CR value is 100%. This index indicates whether the distribution ranges of each variable in the core set are well represented in comparison to the entire collection, and it should be no less than 80% order to represent the whole accessions by core collections [31,51]. MD%, VD%, and VR% are used to measure the statistical consistency between the core and entire collections [31]. If MD% between the initial collection and the core collection is less than 20% of the traits, the VD% within a core collection will be high, indicating the formation of a good core collection [52].

In addition, an ideal core collection encompasses the maximum diversity of the entire germplasm with minimum repetitiveness. Based on these explanations, MD% and VD% of this study were 7.03 and 42.68% respectively, which means a good diversity amongst genotypes within the core collection. A comparison of different values existing in the core collections and the entire collection is calculated by VR%, which determines how well it is being represented in the core sets [31]. The average of this value was 150.2% in the current study, which means that the core collection selected by Power Core is similar to the entire collection.

Calculation of the H′ index in the formed core collection showed that of 25 phenotypic variables, 19 traits had a high diversity with an H′ value of more than 0.67. Four traits, including tree crown shape, flower formation site, fruit ripening time, and aril size had a medium diversity, and fruit skin sensitivity to sunburn and seed hardness had a low diversity. Hence, it should be said that the formed core collection has a good quality with a variety of high attributes.

In this study, the core collection formed with 12 genotypes included 5.4% of the total genotypes of the original collection. In coordination with current results, Yanfang et al. [53] evaluated a collection of 560 mulberry accessions with 40 morphological features in China, declaring that 5% is the best ratio for an ideal core collection in this tree species. However, in the formation of a core collection of olive [54], apricot [20], and apple [55] using molecular markers, collections with a rate of 10–19, 8, and 12.4% were formed respectively, which are higher than that of pomegranate. One reason for these differences is the use of molecular markers that are better able to demonstrate differences among the promising genotypes, resulting in a larger core collection. Also, in a study on 104 Iranian walnut genotypes, a core collection of 27 genotypes, i.e., 26% of the original collection, was formed based on phenotypic diversity [29]. The reason for the higher ratio of walnut core collection to primary collection compared to the results of this study on pomegranate may be due to their different propagation methods. The vegetative propagation of pomegranate trees resulted in less diversity in comparison to walnut trees that are often propagated by seed [56].

Examining the 12 pomegranate superior genotypes in the formed core collection, it was found that these genotypes are equally present in all three population groups formed by cluster analysis. Genotypes, namely "Robab-Ghasrdasht", "Shahvar-Estahban", "Malisak-Hoshak" and "Bazmani-Poost-Koloft" were in C1. Four individuals, "Sheitooni-Seyedon", "Danehsiah-Dastjerd", "Shahvar-poostghermez", and "Ghermez-Harabarjan" were in the C2, and "Sabi-Bam", "Bitalaf-Daneh-Sefid", "Bihasteh-Sangan", and "Poost-Ghermez-Bazman" were classified into the third group.

Production or improvement of cultivars with seedless fruits is one of the main objectives of pomegranate breeding programs. Taking into account that there were only two seedless individuals in the original collection, one of them, i.e., "Bihasteh-Sangan", was included in the constructed core collection. This, in turn, confirm the good quality of the established core collection for breeding objects.

## 5. Conclusions

In this study, as a part of an ongoing project for pomegranate breeding program in Iran, 221 genotypes were clustered into three main groups based on 25 morphological and pomological attributes. Among all of the traits analyzed, those related to flower and fruit had the highest power of discrimination. They are, therefore, the most useful attributes for genetic characterization studies in pomegranate germplasm. The applicable core collection, with 12 representative superior genotypes derived from a whole primary collection, could well represent the Iranian pomegranate variation. These superior genotypes should be used for both production and breeding purposes. To the best of our knowledge, the present study was the first to use phenotypic data for developing a core collection in the pomegranate population. The established core collection will be an efficient step toward exploring, characterizing, and capturing the genetic diversity of large original populations. Also, the conservation of this core collection with only 12 superior genotypes should be more economical and manageable for breeding and production objects. This core collection, in spite of being reduced in size, will provide access to variation quite at the same variation level of initial collection and can be effectively used to improve pomegranate production in Iran and future breeding programs. In order to better evaluate the selected core collection, a complementary characterization of this core collection using molecular markers is also underway.

**Supplementary Materials:** The following are available online at https://www.mdpi.com/article/10.3390/horticulturae7100350/s1, Table S1: Description of the evaluated Iranian pomegranate genotypes consisting of their name and identification code, Table S2: Description of the various phenotypic traits and their frequency in the entire and core collection of the evaluated 221 pomegranate genotypes.

**Author Contributions:** Conceptualization, M.Z.; methodology, S.R. and M.Z.; software, S.R. and M.Z.; validation, M.Z. and A.S.; formal analysis, S.R.; investigation, S.R.; resources, M.R.V.; data curation, M.R.V. and M.T.; writing—original draft preparation, S.R.; writing—review and editing, A.S., M.Z., P.M.-G. and P.J.M.-G.; visualization, S.R. and M.Z.; supervision, A.S. and M.Z.; project administration, M.Z.; funding acquisition, A.M.K. and A.R.R. All authors have read and agreed to the published version of the manuscript.

**Funding:** This work has been supported by Agricultural Biotechnology Research Institute of Iran (ABRII) and partially by Nut4Drought initiative from ARIMNET-2 European Program.

**Conflicts of Interest:** The authors declare no conflict of interest.

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
