# Peer review of "Development of a Multipurpose Core Collection of New Promising Iranian Pomegranate (Punica granatum L.) Genotypes Based on Morphological and Pomological Traits"

_horticulturae, doi:10.3390/horticulturae7100350_

Round 1
Reviewer 1 Report
Dear authors,
The presented article «Development of a multipurpose core collection of new promising Iranian pomegranate (Punica granatum L.) genotypes based on morphological and pomological traits» is devoted to important and topical problems, which have a pronounced applied value for plant breeding. This article is of great interest to the world scientific community. However, unfortunately, there are both insignificant and significant remarks, as well as issues that should be taken into account.
- In my opinion, the INTRODUCTION section needs to be expanded, since it is written superficially. For example, what are the main problems and breeding programs relevant to pomegranate? What are the main morphological and pomological characteristics used for pomegranate and how do they correlate with economically valuable traits? How widely are these characteristics used in pomegranate breeding?
- In this work, the authors estimate various indicators for 2 years (2012-2013). Why were the research results evaluated specifically for 2012-2013? Did these years differ significantly in terms of climatic conditions? Were the growing conditions typical or contrasting for the region? This question remained unclear for me. It is necessary to supplement this section with data, for example, on the average monthly air temperature for the growing season in comparison with average annual values, as well as data on the amount of precipitation for 2012 and 2013.
- It is not clear why the authors chose these 25 morphological and pomological characteristics? This needs to be argued.
- The authors extremely poorly placed figures and tables separately from the text, as well as their captions. This makes the review process very difficult.
- Most of the figures are extremely difficult to read. This is especially true for Fig. 3 and 6. Unfortunately, I could not draw any conclusions about the correctness of their interpretation.
Best regards, reviewer
Reviewer 2 Report
The manuscript deals with a relevant subject to Horticulturae, related with germplasm phenotypic diversity of pomegranate. The ms is very interesting, well written, with an attractive set of appropriate topics, well supported by the literature. I recommend that the manuscript should be accepted after minor revision.
Specific comments
- Line 126: some examples of morphological attributes should be provided.
- Line 131: some examples of intermediate and low variability should be provided.
- Line 229: anti-transpiration instead of anti-transpiring.
Reviewer 3 Report
In this manuscript, authors established a core collection of pomegranate genotype based on 25 morphological traits. The study might be interested for someone who are working on pomegranate breeding. However, the manuscript does have several issues which I think need to be addressed.
- Maintaining genetic diversity is essential to the long term breeding program and normally requires a large number of collection of genotypes. In plant material, 221 out of 760 genotypes were selected before this study. Why do authors think that it is necessary to further select down to 12 out of 221 as a core population? Is 12 genotypes enough for the core collection in breeding program?
- A total of 25 traits related to tree, leaf, flower, and fruit were measured. How does environment affect these traits, or do phenotype traits behave differently in different year? Have authors ever consider to use traits related to cold hardness and disease resistance?
- Using morphological traits alone might not be enough to establish core germplasm diversity, since phenotype variation might not represent genotype variation. Why don't authors consider to use molecular markers or combine information of molecular markers together with phenotype traits to establish core collection?
- Some descriptors only have two or three outcomes, such as TCS, LWLR, FSC,FCS, FSY. Why do authors use four or five grading scores, more than the number of outcomes?
Round 2
Reviewer 1 Report
Dear authors,
Thank you for taking into account all the comments and remarks regarding the article. I am sure it was helpful and the quality of the article has improved. The article "Development of a multipurpose core collection of new promising Iranian pomegranate (Punica granatum L.) genotypes based on morphological and pomological traits" can be accepted for publication in Horticulturae journal.
Reviewer 3 Report
Authors addressed my comments. I think the paper is now acceptable for publication.